# Antibody and Cellular Immune Responses in Old α1,3-Galactosyltransferase-Knockout Mice Implanted with Bioprosthetic Heart Valve Tissues

**DOI:** 10.3390/bioengineering13010053

**Published:** 2025-12-31

**Authors:** Kelly Casós, Roger Llatjós, Arnau Blasco-Lucas, Sebastián G. Kuguel, Fabrizio Sbraga, Cesare Galli, Vered Padler-Karavani, Thierry Le Tourneau, Marta Vadori, Jean-Christian Roussel, Tomaso Bottio, Emanuele Cozzi, Jean-Paul Soulillou, Manuel Galiñanes, Rafael Máñez, Cristina Costa

**Affiliations:** 1Infectious Diseases and Transplantation Division, Institut d’Investigació Biomèdica de Bellvitge (IDIBELL), L’Hospitalet de Llobregat, 08908 Barcelona, Spain; kcasos@recerca.clinic.cat (K.C.);; 2Department of Pulmonary Medicine, Servei de Pneumologia, Hospital Clínic-Institut d’Investigacions Biomèdiques August Pi I Sunyer (IDIBAPS), University of Barcelona, 08036 Barcelona, Spain; 3Pathology Department, Bellvitge University Hospital, L’Hospitalet de Llobregat, 08907 Barcelona, Spain; 4Cardiac Surgery Department, Bellvitge University Hospital, L’Hospitalet de Llobregat, 08907 Barcelona, Spain; 5Bioengineering Department, Institut Químic de Sarrià (IQS), Universitat Ramon Llull (URL), 08017 Barcelona, Spain; 6Avantea, Laboratory of Reproductive Technologies, Via Porcellasco 7/F, 26100 Cremona, Italy; 7Department of Cell Research and Immunology, The Shmunis School of Biomedicine and Cancer Research, The George S. Wise Faculty of Life Sciences, Tel Aviv University, Tel Aviv 69978, Israel; vkaravani@tauex.tau.ac.il; 8Institut du Thorax, INSERM UMR1087, Nantes University Hospital, 8 Quai Moncousu, 44007 Nantes, France; 9Transplantation Immunology Unit, Padua University Hospital, 35128 Padova, Italy; 10Department of Cardiac, Thoracic, Vascular Sciences and Public Health, University of Padova Medical School, 35122 Padova, Italy; 11Institut de Transplantation–Urologie–Néphrologie, INSERM Unité Mixte de Recherche 1064, Nantes University Hospital, 44093 Nantes, France; 12Department of Cardiac Surgery and Reparative Therapy of the Heart, Vall d’Hebron Research Institute (VHIR), University Hospital Vall d’Hebron, Universitat Autònoma de Barcelona, 08035 Barcelona, Spain; 13Intensive Care Department, Bellvitge University Hospital, L’Hospitalet de Llobregat, 08907 Barcelona, Spain; 14Institut de Química i Biotecnologia de Barcelona (IQBB), Consorci d’Educació de Barcelona (CEB), 08003 Barcelona, Spain

**Keywords:** bioprosthetic heart valves, structural valve deterioration, aging, α1,3-galactosyltransferase-knockout mice, anti-Gal antibodies, cellular immune response

## Abstract

Structural valve deterioration (SVD) remains a key limitation in bioprosthetic heart valve (BHV) usage influenced by patient age. A deeper understanding of SVD pathogenesis, particularly of the immune-mediated processes altering current BHV materials, is therefore critical. To this end, commercially available BHV tissues of bovine, porcine, and equine origin were investigated following subcutaneous implantation into α1,3-galactosyltransferase-knockout (Gal KO) mice. We compared the immune responses between adult and aged animals via histological assessments of explants and measurement of serum anti-galactose α1,3-galactose (Gal) and anti-non-Gal antibodies at 2 months post-implantation. In contrast to adult mice, old Gal KO mice did not show increased levels of serum anti-Gal or -non-Gal antibodies after receiving specific BHV tissue (i.e., Freedom-Solo). Instead, a significant decrease in serum anti-Gal IgM was found in old recipients of Freedom-Solo. Furthermore, the overall cellular immune response was attenuated in explants from old mice compared with adults (i.e., ATS 3f and Crown). Nevertheless, the Freedom-Solo (bovine) and the Hancock-II (porcine) tissues still elicited strong cellular immune infiltration in the old cohorts. Therefore, the Gal KO mouse model offers a valuable platform to investigate age-related differences regarding cellular and humoral immune responses to various BHV tissues, contributing to our understanding of SVD.

## 1. Introduction

The increased life expectancy and the aging of the global population have led to a growing clinical demand for bioprosthetic heart valves (BHVs) [1,2,3]. BHVs are commercially developed for surgical heart-valve replacement [4,5] and transcatheter aortic valve implantation (TAVI) [6,7]. In addition, the BHV processed tissues are also incorporated into other cardiac devices [8,9]. A major limitation to the broader adoption of BHVs, particularly in surgical valve replacement, is structural valve deterioration (SVD) [1,2,3,4,5,10,11,12,13,14,15]. SVD results from the loss of tissue integrity and calcification of the processed biomaterials driven by a multifactorial pathogenesis [10,11,12,13,14,15]. Of particular interest, immune responses to the implanted BHV tissues contribute to SVD, yet the mechanisms remain incompletely understood [11,12,13,14]. Therefore, investigating the immunogenicity of BHVs—both those currently used and those under development—is essential for predicting their durability and clinical performance.

A primary source of BHV immunogenicity lies in their xenogeneic origin, as these valves are derived from bovine or porcine pericardium, porcine valves, and in some cases, equine pericardium [4,11,12,13,14,16,17]. These animal tissues express carbohydrates such as the galactose α1,3-galactose (Gal) epitope and the N-glycolylneuraminic acid (Neu5Gc) antigen, which are recognized by specific human natural antibodies [16,17,18]. These antigens are not fully eliminated during the BHV manufacturing process and elicit antibody responses that contribute to calcification and SVD [13]. Furthermore, glutaraldehyde fixation, a common method used to reduce protein immunogenicity, may paradoxically promote calcification and SVD [4,12]. As a result, anti-calcification treatments are routinely incorporated into the preparation of commercial BHV tissues. Beyond humoral immunity, cellular immune responses have also been found in BHV explanted from patients with SVD [11,19,20]. These immune responses have been reproduced in subcutaneous implantation models using α1,3-galactosyltransferase knockout (Gal KO) mice, where both cellular and antibody-mediated responses act independently, determining the outcome of the specific commercial tissue tested [21]. The use of this mouse model is highly relevant to study immune responses in a xenogeneic setting. Like humans, these mice produce anti-Gal antibodies, and they develop a stronger cellular immune response to Gal-bearing tissues than wild-type mice [21,22,23]. The immune responses to carbohydrates show particularities and the Gal KO mice could be considered for modeling anti-carbohydrate immune responses beyond the Gal antigen [22].

Patient age is a key factor influencing BHV durability and serves as major criterion for selecting mechanical heart valves (MHVs) over BHVs [2,3,10,24]. Although it is presumed that the strength of immunity plays a role, very few studies have directly examined the impact of immunological aging on SVD. Furthermore, the mechanisms that contribute to early SVD in younger patients remain poorly understood [12,25]. Hence, studies characterizing these immunological processes are highly justified.

In this work, the humoral and cellular immune responses to commercially available BHV tissues were compared in old versus adult mice using the subcutaneous Gal KO mouse model. We observed a reduction in both antibody and cellular responses in old mice that differed depending on the BHV tissue studied.

## 2. Materials and Methods

### 2.1. Obtention and Processing of Tissues

Medtronic and Sorin-LivaNova kindly donated BHVs for this study. The BHVs were stored in their original packaging until use. The human pericardium was obtained from a patient undergoing heart surgery, and it was subjected to fixation in 0.6% glutaraldehyde (Merck KGaA, Darmstadt, Germany) in phosphate-buffered saline (PBS, Merck) for one month, followed by maintenance in 0.2% glutaraldehyde. In preparation for surgical implantation, small tissue fragments from BHVs and the human pericardium were prepared under conditions of sterility, weighted, and extensively rinsed in tissue culture-grade PBS. The processed samples were then incubated overnight in PBS in 24-well plates. All procedures involving human tissue were approved by the Clinical Research Ethics Committee of Bellvitge University Hospital in accordance with current legislation.

### 2.2. Mouse Model of Subcutaneous BHV Implantation

All animal practices complied with guidelines of the European Commission. The protocols were submitted and approved by the local institutional ethical committee and Generalitat de Catalunya. The α1,3-galactosyltransferase knockout (Gal KO) mice utilized for this study belong to our in-house colony, have a mixed background (B6 × CBA × 129sv), and typically produce natural anti-Gal antibodies after 4 months of age [21,23]. Mice 18–22 months old were included in the study and evenly distributed across experimental groups (*n* = 7–13 per group of old mice), comprising both sexes and a range of pre-determined anti-Gal IgM and IgG antibody titers, quantified by ELISA using Gal-conjugated human serum albumin (Gal-HSA)-coated plates, as described [23]. The experimental conditions for the adult cohorts (10–14 months old) used for comparison were previously described [21].

Five different experimental cohorts were established to assess the immune responses to four commercially available BHVs: ATS 3f, Crown (same material as Mitroflow PRT), Freedom Solo, or Pericarbon Freedom (both with the same tissue, referred to as “Freedom”), and Hancock II, and a fifth cohort for the glutaraldehyde-fixed human pericardium. Tissue samples (7–10 mg wet weight) were matched across groups, and a single piece was implanted subcutaneously into the dorsal region of each mouse under isoflurane anesthesia, maintaining sterile conditions. Body weight and blood samples were collected at baseline and again at terminus, two months post-implantation. Sera were isolated from clotted blood and stored frozen for subsequent analyses of xenoantibodies.

### 2.3. Histopathological Analysis

The explanted grafts were fixed in formalin (Merck), processed, and then embedded in paraffin for histological examination. Tissue sections (3 µm) were stained with hematoxylin and eosin (H&E), following standard protocols, and assessed independently by two investigators in a blinded manner. A semi-quantitative scoring system (scale 1–5) was applied to assess the degree of graft infiltration by immune cells, as described in figure legends. The extent of preserved BHV tissue was quantified in representative 100×-magnification images using the software of ImageJ.JS (https://ij.imjoy.io/, U.S. National Institutes of Health, Bethesda, MD, USA) by delineating total and preserved tissue areas and calculating the percentage of tissue preserved.

### 2.4. Culture of Endothelial Cells

The European Collection of Cell Cultures (Porton Down, Salisbury, UK) was the source of porcine aortic endothelial cells (PAECs), which were seeded in culture flasks (TPP Techno Plastic Products AG, Trasadingen, Switzerland) pre-coated with 1% porcine collagen (Merck) in PBS to support cell adhesion and growth. PAECs were cultured in Dulbecco’s Modified Eagle’s Medium (DMEM) containing 10% fetal bovine serum (FBS) and supplemented with 100 IU/mL penicillin–100 μg/mL streptomycin and 50 μg/mL endothelial cell growth supplement (Millipore, Burlington, MA, USA, Merck).

### 2.5. Determination of Xenoantibodies in Serum

An in-house flow cytometry-based method was used to assess total xenoantibody responses, involving both anti-Gal and anti-non-Gal antibodies [21]. PAECs grown to confluency were harvested with TripLE Express (Thermo Fisher Scientific, Waltham, MA, USA), washed twice, and incubated at 4 °C for 30 min with mouse sera (1% or 0.5%) in PBS containing 1% bovine serum albumin (Merck), either alone or with 0.5 mg/mL of GAS914 (Novartis, Basel, Switzerland), which blocks anti-Gal antibodies [26,27]. IgM and IgG reactivities were detected using Alexa Fluor 647-conjugated goat anti-mouse IgM (1:200) and PE-conjugated goat anti-mouse IgG (1:150) secondary antibodies (both from Thermo Fisher Scientific, Waltham, MA, USA). The values of mean fluorescence intensity (MFI) were obtained for each mouse sample (including all time points) in a Gallios flow cytometer (Beckman Coulter, Brea, CA, USA), using Kaluza Analysis 2.1 software (Beckman Coulter), after subtracting the background. Background was established with reactivities obtained in the presence of GAS914 for the anti-Gal antibodies, whereas the residual reactivity following GAS914 competition over that of secondary antibody alone was attributed to anti-non-Gal antibodies. The only gating applied during the analyses was the selection of a homogeneous PAEC population in the SSC-FSC dot plot. A more detailed explanation of this procedure, including representative original images of the flow cytometry data, is presented in the Appendix A.

### 2.6. Statistical Analyses

GraphPad Prism 6 (GraphPad Software, Boston, MA, USA) was utilized for statistical analyses. Data are displayed as mean ± standard error of the mean (SEM) when indicated. First, data normality was assessed using the Shapiro–Wilk test. For normally distributed data, comparisons were made using one-way ANOVA and Tukey’s test for multiple comparisons, one-way ANOVA and Dunnett’s post hoc tests, and paired Student’s *t*-test for individual comparisons. For non-normally distributed data (e.g., some studies involving old mice), the Kruskal–Wallis with Dunn’s tests for multiple comparisons or the Mann–Whitney U test/Wilcoxon signed-rank test for single comparisons was applied. Correlations were evaluated using the coefficient of determination (R^2^). Significance was set at *p* ≤ 0.05.

## 3. Results

### 3.1. Gal KO Mice Develop an Age-Dependent Cellular Immune Response to Specific BHV Tissues

The immune response to BHV tissues has been previously characterized in young and adult Gal KO mice [21,28,29]. In this study, we investigated whether aging influences the immune response to selected commercially available BHV tissues in the Gal KO mouse model. To this end, commercial BHV tissues of equine, bovine, and porcine origin, as well as glutaraldehyde-fixed human pericardium, were implanted subcutaneously in old Gal KO mice. After two months, the implants were explanted and subjected to histological analysis (Figure 1). Results were compared with previously published data from adult Gal KO mice of the same in-house colony [21]. Whereas adult mice exhibited stable body weights at 2 months post-implantation, old mice showed significant weight loss across most experimental groups at this time point (Table 1).

A histological analysis of sections of the BHV tissues after explantation was conducted as previously described [21]. It involved H&E staining and a semi-quantitative scoring (1–5) of cellular infiltration to determine the local immune response to the BHV tissue. Most explanted tissues exhibited cellular infiltrates originating from the graft periphery (Figure 2a). Among old Gal KO mice, the strongest cellular immune responses were found in the Freedom and the Hancock-II tissues, followed by the glutaraldehyde-fixed human pericardium (Figure 2a,b). In contrast, the ATS 3f graft sections triggered a mild cellular immune response, while the Crown showed the weakest cellular immune response overall. These findings were corroborated by immune response scores, which were significantly higher for Freedom and Hancock II than for ATS 3f and Crown (Figure 2b). Interestingly, the proportion of myeloid and lymphoid cells in the cellular infiltrates, as determined in the histologic analysis (Figure 2a), appeared comparable to that previously reported for adult mice [21]. Lastly, a measure of tissue preservation (displayed in the Figure 2 legend) was determined for the selected samples with results that paralleled the histology scores of cellular immune responses. Thus, the highest degree of structural preservation, relative to the 100% integrity of non-implanted controls [21], corresponded to the Crown explants, followed by the ATS 3f.

Next, a comparative analysis of the scoring results from adult and old cohorts was conducted (Table 2). The direct comparison revealed a significant attenuation of the cellular immune response for ATS 3f, Crown, and human pericardium associated with older age. However, no such reduction occurred in the response to Freedom or Hancock II tissues explanted from old Gal KO mice compared with the adult cohorts (Table 2), suggesting that some BHV tissues retain their immunogenicity even in the context of immunosenescence.

### 3.2. The Antibody Response to Specific BHV Tissues Is Defective in Old Gal KO Mice

To evaluate the humoral immune response, we analyzed anti-Gal and anti-non-Gal antibody serum levels in adult and old Gal KO mice after BHV implantation using a previously established flow cytometry method [21] (Table 3, Table 4, Table 5 and Table 6). Previous work in adult Gal KO mice demonstrated that implantation of BHV tissues such as ATS 3f and Freedom leads to a significant induction of anti-Gal antibodies at 2 months post-implantation [21].

Anti-Gal IgM data are summarized in Table 3. In adult Gal KO mice, the Freedom tissue induces a robust and statistically significant increase in anti-Gal IgM reactivity (Table 3). ATS 3f also elicited increased anti-Gal IgM in most recipients, though not reaching statistical significance. Strikingly, old Gal KO mice implanted with the same BHV tissues did not replicate these antibody responses (Table 3 and Figure 3). Specifically, Freedom recipients of old age exhibited a significant reduction in anti-Gal IgM levels compared to their baseline (before implantation), and most ATS 3f recipients (8 out of 13) similarly showed reduced titers (Figure 3). However, a subset of ATS 3f recipients (4 out of 13) exhibited increased anti-Gal IgM levels, thereby precluding a statistically significant overall reduction.

Although the anti-Gal IgG responses were less pronounced than those of IgM, these were consistent with a general reduction of humoral immunity in old mice (Table 4 and Figure 3). Notably, the increase observed in anti-Gal IgG in adult ATS 3f recipients was absent in the old cohort. Overall, these findings indicate a defective anti-Gal B cell response in old Gal KO mice.

**Table 3 bioengineering-13-00053-t003:** Serum anti-Gal IgM reactivity (mean ± SEM of mean fluorescence intensity, MFI) of adult and old Gal KO mice grafted subcutaneously with BHV tissues for 2 months ^1^.

Anti-Gal IgM	Adult Mice		Old Mice	
BHV	Baseline	2 months	Baseline	2 months
ATS 3f	17.0 ± 3.32	21.8 ± 5.86	130.1 ± 28.46	104.5 ± 29.76
Crown	16.1 ± 3.63	13.3 ± 4.81	35.1 ± 6.96	43.8 ± 11.56
Freedom	19.6 ± 5.24	28.6 ± 6.05 *	137.4 ± 31.26	55.2 ± 11.72 *
Hancock	26.7 ± 13.74	25.9 ± 7.27	36 ± 8.14	30.9 ± 8.77
huPeric	19.6 ± 4.44	13.4 ± 2.63	74.5 ± 16.99	59.6 ± 10.94

^1^ Adult Gal KO mice were implanted with fragments of either ATS 3f (*n* = 8), Crown (*n* = 9), Freedom Solo (Freedom, *n* = 9), Hancock II (*n* = 7), or human pericardium (huPERIC, *n* = 7), and the anti-Gal IgM antibodies were determined by FACs, as described. Old Gal KO mice also received either ATS 3f (*n* = 13), Crown (*n* = 9), Freedom Solo (Freedom, *n* = 9), Hancock II (*n* = 9), or human pericardium (huPERIC, *n* = 8), and the anti-Gal IgM was equally determined by FACs. Statistical significance relative to baseline was determined by paired Student’s *t*-test for both adult mice and old mice (* *p* ≤ 0.05). Note that experimental conditions do not allow us to directly compare the values of adult and old mice.

**Table 4 bioengineering-13-00053-t004:** Serum anti-Gal IgG reactivity (mean ± SEM of mean fluorescence intensity, MFI) of adult and old Gal KO mice grafted subcutaneously with BHV tissues for 2 months ^1^.

Anti-Gal IgG	Adult Mice		Old Mice	
BHV	Baseline	2 months	Baseline	2 months
ATS 3f	1.47 ± 0.39	1.84 ± 0.49 *	1.55 ± 0.48	1.77 ± 0.49
Crown	2.43 ± 0.73	2.16 ± 0.85	1.21 ± 0.43	0.68 ± 0.25
Freedom	3.69 ± 1.22	3.56 ± 1.03	1.53 ± 0.43	2.56 ± 0.89
Hancock	3.52 ± 1.53	4.05 ± 1.37	0.86 ± 0.30	2.56 ± 0.89
huPeric	0.92 ± 0.38	1.29 ± 0.63	1.41 ± 0.31	1.02 ± 0.26

^1^ Adult Gal KO mice were implanted with fragments of either ATS 3f (*n* = 8), Crown (*n* = 9), Freedom Solo (Freedom, *n* = 9), Hancock II (*n* = 7), or human pericardium (huPERIC, *n* = 7), and the anti-Gal IgG antibodies were determined by FACs, as described. Old Gal KO mice also received either ATS 3f (*n* = 13), Crown (*n* = 9), Freedom Solo (Freedom, *n* = 9), Hancock II (*n* = 9), or human pericardium (huPERIC, *n* = 8), and the anti-Gal IgG was equally determined by FACs. Statistical significance relative to baseline was determined by paired Student’s *t*-test for both adult mice and old mice (*******
*p* ≤ 0.05). Note that experimental conditions do not allow us to directly compare the values of adult and old mice.

Regarding anti-non-Gal antibodies, old mice exhibited no meaningful changes in IgM or IgG responses (Table 5 and Table 6). In adult mice, a mild but significant anti-non-Gal IgG response had previously been observed only in Freedom recipients [21]; this response was not detectable in old mice (Table 6). Furthermore, we found no differences in the reactivities (Table 3, Table 4, Table 5 and Table 6) and overall ratios of baseline anti-Gal and anti-non-Gal antibodies between the adult and old Gal KO cohorts (2.6 and 2.7 for IgM, and 1.66 and 1.5 for IgG in adult and old groups, respectively). Altogether, our data indicated that the humoral immunity defects in old Gal KO mice mainly affected the elicited antibody response without compromising the generation of natural antibodies.

**Table 5 bioengineering-13-00053-t005:** Serum anti-non-Gal IgM reactivity (mean ± SEM of mean fluorescence intensity, MFI) of adult and old Gal KO mice grafted subcutaneously with BHV tissues for 2 months ^1^.

Anti-Non-Gal IgM	Adult Mice		Old Mice	
BHV	Baseline	2 months	Baseline	2 months
ATS 3f	5.2 ± 0.67	6.4 ± 0.72	29.8 ± 2.88	31.7 ± 5.36
Crown	5.7 ± 0.88	8.3 ± 1.79	45.2 ± 7.05	36.5 ± 6.76
Freedom	5.4 ± 0.81	6.4 ± 1.07	41.4 ± 8.08	43.4 ± 10.24
Hancock	20.4 ± 4.67	22.7 ± 5.51	29.8 ± 7.53	29.6 ± 8.62
huPeric	9.3 ± 2.43	10.5 ± 3.04	18.4 ± 2.90	20.3 ± 2.83

^1^ Adult Gal KO mice were implanted with fragments of either ATS 3f (*n* = 8), Crown (*n* = 9), Freedom Solo (Freedom, *n* = 9), Hancock II (*n* = 7), or human pericardium (huPERIC, *n* = 7), and the anti-non-Gal IgM antibodies were determined by FACs, as described. Old Gal KO mice also received either ATS 3f (*n* = 13), Crown (*n* = 9), Freedom Solo (Freedom, *n* = 9), Hancock II (*n* = 9), or human pericardium (huPERIC, *n* = 8), and the anti-non-Gal IgM was equally determined by FACs. No statistical significance relative to baseline was observed as determined by paired Student’s *t*-test for both adult mice and old mice (*p* ≤ 0.05). Note that experimental conditions do not allow us to directly compare the values of adult and old mice.

**Table 6 bioengineering-13-00053-t006:** Serum anti-non-Gal IgG reactivity (mean ± SEM of mean fluorescence intensity, MFI) of adult and old Gal KO mice grafted subcutaneously with BHV tissues for 2 months ^1^.

Anti-Non-Gal IgG	Adult Mice		Old Mice	
BHV	Baseline	2 months	Baseline	2 months
ATS 3f	1.87 ± 0.32	1.80 ± 0.33	0.64 ± 0.15	0.64 ± 0.24
Crown	1.79 ± 0.65	2.32 ± 1.25	0.87 ± 0.27	0.81 ± 0.19
Freedom	1.18 ± 0.22	1.49 ± 0.19 **	0.70 ± 0.21	0.96 ± 0.44
Hancock	1.92 ± 0.37	3.70 ± 1.03	1.09 ± 0.39	1.89 ± 0.53
huPeric	2.62 ± 0.97	2.31 ± 0.88	1.67 ± 0.34	2.36 ± 0.54

^1^ Adult Gal KO mice were implanted with fragments of either ATS 3f (*n* = 8), Crown (*n* = 9), Freedom Solo (Freedom, *n* = 9), Hancock II (*n* = 7), or human pericardium (huPERIC, *n* = 7) and the anti-non-Gal IgG antibodies were determined by FACs as described. Old Gal KO mice also received either ATS 3f (*n* = 13), Crown (*n* = 9), Freedom Solo (Freedom, *n* = 9), Hancock II (*n* = 9), or human pericardium (huPERIC, *n* = 8), and the anti-non-Gal IgG was equally determined by FACs. Statistical significance relative to baseline was determined by paired Student’s *t*-test for both adult mice and old mice (** *p* ≤ 0.005). Note that experimental conditions do not allow us to directly compare the values of adult and old mice.

### 3.3. Limited Correlation Between the Anti-Gal Antibodies and the Cellular Immune Response Elicited by BHVs in Old Gal KO Mice

To explore whether cellular and humoral immune responses to BHV tissues were interrelated, we looked at correlations between the anti-Gal antibody levels and the histological scores in old mice (Figure 4). Previous studies in adult Gal KO mice found no significant correlation between graft infiltration of immune cells and serum anti-Gal antibodies at various time points and across BHV types [21]. In the present study, most BHV tissues again showed no correlation between these parameters (Figure 4). However, two notable exceptions were identified: (1) in Freedom recipients, a significant positive correlation was observed between histological scores and anti-Gal IgM levels at 2 months post-implantation; and (2) in ATS 3f recipients, the degree of tissue infiltration significantly correlated with elicited anti-Gal IgG levels (Figure 4). These findings suggest that this dependency of the cellular and antibody effects can only be detected in the context of immunosenescence.

## 4. Discussion

In this study, we provide evidence, using a relevant mouse model, that the advanced recipient age leads to diminished cellular and antibody responses to several commercially available BHV tissues. To the best of our knowledge, this is the first use of an experimental animal model to directly investigate the effect of aging on the immune response to BHV implants. Our findings show that both recipient age and the specific BHV tissue influence the strength of the immune reaction to the BHV graft and, ultimately, graft outcome.

Age is a well-established modifying factor for SVD [3,10]. In surgical valve replacement, whether aortic or mitral, SVD progression is slower in older patients [3,10]. Thus, the risk for developing SVD after surgical aortic valve replacement is significantly higher in patients under 60 years compared to older individuals [10]. Likewise, a large meta-analysis of surgical mitral valve replacement found greater freedom from SVD in patients over 70 than in those aged 18–59 [2]. Notably, Goldstone et al. identified an age threshold below which surgical BHV implantation led to worse survival than with MHV (under 55 years old for aortic valve replacement and under 70 years for the mitral position) [24] Our Gal KO mouse results align with a protective effect of older age, potentially due in part to a diminished immune response to the implanted BHV. Considering our data, we discuss below possible underlying mechanisms.

Regarding cellular immunity, old Gal KO mice exhibited reduced cellular immune infiltration compared to the adult cohorts for several of the BHV tissues studied (ATS 3f, Crown, and human pericardium). This mirrors clinical observations of longer BHV durability in elderly patients [3,10,24] and is consistent with immunosenescence in mice and humans [30,31,32,33,34,35]. In BHV explants from both patients and mice, monocytes/macrophages are the predominant immune cells, followed by T cells [11,19,20,21]. With age, both cell types are subjected to modifications that are overall associated with increased basal cytokine production but reduced cellular effector functions upon activation [30,31,32,33,34]. Aged monocytes produce less type I interferons and exhibit restricted efferocytosis [30,31,32]. Aged macrophages show instead reduced Toll-like receptors and impaired capabilities in chemotaxis, phagocytosis, and inflammation resolution [30,31,32,33]. T-cell changes are a hallmark of immunosenescence [33,34,35]. Aging reduces naïve T-cell numbers; effector and memory T cells accumulate; T-cell receptor diversity declines; and CD28 expression—essential for activation and proliferation—is lost [33,34,35]. All of these processes could affect the cellular immune responses to BHV tissues in old Gal KO mice and in patients.

Our findings suggest that chemical treatments such as glutaraldehyde may be particularly effective at reducing immunogenicity of certain BHV tissues (i.e., ATS 3f and Crown) in older recipients by limiting monocyte and T-cell recruitment and activation. Conversely, BHVs like Freedom Solo and Hancock II still triggered robust cellular immune responses in old mice. These results are consistent with the strong immunogenicity of these tissues previously observed in adult Gal KO mice [21] and suggest that older mice may have difficulties in controlling the stronger inflammatory responses triggered by these xenogeneic tissues. The clinical implications for SVD are uncertain but deserve further study.

Clear age-related differences were also observed in antibody responses in the Gal KO mouse model. Old Gal KO mice showed no elicited anti-Gal and anti-non-Gal antibody responses after BHV implantation, whereas the grafting of some of these BHV tissues (Freedom Solo and ATS 3f) induced an antibody response in adult mice under equivalent experimental conditions. Older Freedom recipients exhibited instead a significant reduction in serum anti-Gal IgM reactivity, likely due to antibody absorption by the graft. Several points merit consideration. (1) Clinical parallels: Data in humans are limited. Park et al. found higher baseline anti-Gal IgM and IgG in patients younger than 65 years versus older patients, but not post-implantation with this age stratification [36]. A higher threshold, such as 70, and larger cohorts may reveal differences. (2) Antibody absorption: Reductions in serum anti-Gal IgM and IgG shortly after BHV implantation in patients have been reported [36] and are consistent with graft antibody absorption, as in xenotransplantation models (i.e., after pig heart transplantation in non-human primates) [37]. Moreover, anti-Gal antibodies have been detected in BHVs obtained from patients [13]. (3) Immunosenescence effects: Reduced antibody responses in old Gal KO mice implanted with BHVs likely reflect age-related changes in both B-cell and T-cell functions. In particular, the defects could be related to B-cell maturation, class switching, and somatic hypermutation, alongside T-cell and antigen-presenting-cell alterations [33,34]. As a limitation of this study, baseline anti-Gal levels were not directly compared between age groups, but pre-implantation reactivities and anti-Gal:anti-non-Gal ratios were similar.

Regarding antibody-cellular interactions, specific conditions yielded positive correlations between serum antibody reactivity and histological cellular scores in old Gal KO mice, particularly for anti-Gal IgM and IgG at two months. This finding contrasted with the lack of such correlations in adults [21]. Mechanistically, IgM binding activates complement, generating anaphylatoxins that recruit immune cells [38,39,40], potentially explaining persistent cellular immune responses in Freedom recipients. Notably, this BHV tissue elicits anti-Gal IgM in adult mice [21]. IgG can similarly activate complement and engage Fcγ receptors to enhance cellular immunity [38,41], which may explain responses in ATS 3f recipients. Additionally, ATS 3f induces anti-Gal IgG in adults [21]. Interestingly, our observations agree with clinical results in elderly patients after TAVI, who show increases in both anti-Gal IgG3 and blood markers of inflammation 3 months after implantation [42]. Overall, these data suggest that antibody and cellular responses to BHVs may operate independently and in concert. It is likely that deficiencies in immune responses in old mice allow for the visualization of these positive correlations and thus antibody–cellular interactions.

Although overall immune responses were weaker in old mice, the type of response to each BHV tissue was consistent with that in adults. Hancock II (porcine valves) elicited a strong cellular immunity in adult and old Gal KO mice, and the pericardial BHV tissues, Freedom and ATS 3f, produced mixed humoral and cellular responses in both age groups. These patterns reinforce the concept that BHV-specific immune profiles influence clinical outcomes, such as tears, calcification, and fibrosis [21,43,44]. Direct implications are expected, as many patients wear the BHVs here studied, and the Perceval valve (with the same tissue as the Freedom BHVs in the first product) is highly grafted in the elderly [45].

This study focuses on one of the various carbohydrate antigens found on xenogeneic tissues known to be absent in humans and trigger immune reactions [13,18,46,47,48]. The Gal KO mouse could be considered as a model to study all of them, as it is well characterized. Nevertheless, it may be of interest in the future to conduct similar studies in mice deficient for Neu5Gc, SD(a), or combinations with Gal KO to assess the impact of other xenogeneic carbohydrates on BHVs. The flow cytometric method developed to assess the anti-Gal and anti-non-Gal antibody responses may be of use in such studies after subjected to the necessary adaptations.

Regarding the feasibility of using old Gal KO mice, these were monitored visually and by body weight measurements. The protocol was modified from that for adult mice to minimize handling (blood was only collected at the beginning and end of the study). No losses occurred, though most cohorts experienced some reduction in body weight, likely age-related, as it was absent in adults. A key aim was to test the feasibility of using old Gal KO mice for modeling elderly patients. Despite the greater time and cost to obtain old mice, this first study demonstrates their preclinical value. Protocols can be adapted for each strain and colony to allow for research to be performed on old mice without compromising their welfare.

## 5. Conclusions

Our findings support the use of the Gal KO mouse model across age ranges to study the immune responses to BHV and other xenogeneic tissues. In older mice, overall immunity was reduced compared with their younger cohorts, but distinct tissue-specific patterns persisted—remaining strong for certain BHVs. Characterizing these age- and tissue-dependent differences may guide the selection and design of BHVs optimized for different patient populations.

## Figures and Tables

**Figure 1 bioengineering-13-00053-f001:**
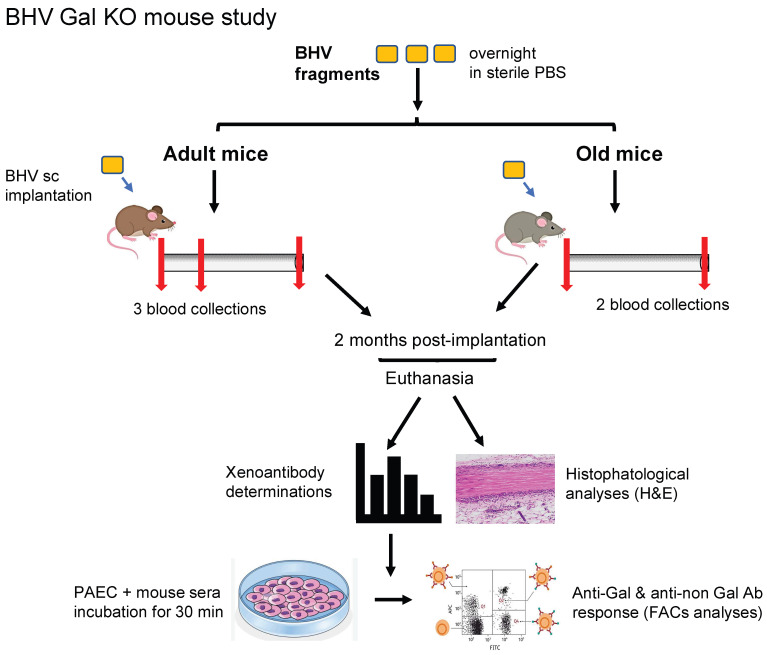
Experimental design of Gal KO mice implanted subcutaneously with commercial BHV tissues for 2 months. A scheme of the experimental procedure is shown that includes a description of the procedure for both adult and old Gal KO mice. Selected BHV fragments were washed and incubated in PBS overnight prior to implantation. A single piece was grafted subcutaneously onto the dorse of each experimental mouse and kept for 2 months prior to explantation for histological analysis. Blood was collected (red arrows) before and at the end of the experiment for all mice, as well as at 3 weeks post-implantation in adult mice, for serum obtention and xenoantibody determinations by flow cytometry, as described in the Materials and Methods section.

**Figure 2 bioengineering-13-00053-f002:**
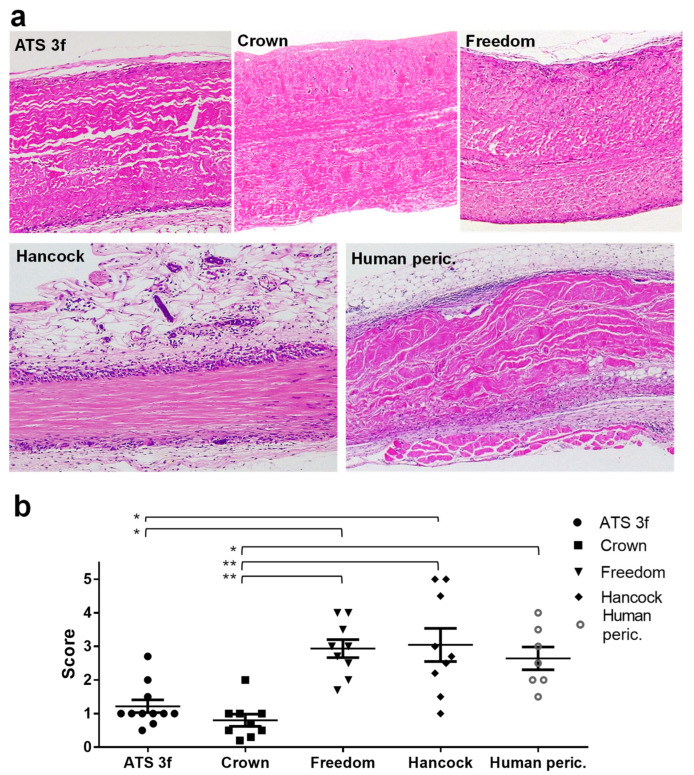
Histological analysis of BHV tissues explanted from old Gal KO mice at 2 months post-implantation. (**a**) Results of H&E staining of paraffin sections of the indicated BHVs are shown as representative images. Original magnification: ×100. The percentage of tissue preservation calculated for the representative images is 88.3% for ATS 3f, 97.5% for Crown, 85.4% for Pericarbon Freedom/Freedom Solo (Freedom), 65.05% for Hancock II (Hancock), and 65.1% for human pericardium (Human peric.). (**b**) Individual data and mean ± SEM of scores assigned to describe the amount of cellular immune infiltrate (low = 1, modest = 2, medium = 3, high = 4, and very high = 5) for the analyzed BHV grafts (ATS 3f (*n* = 11), Crown (*n* = 9), Freedom (*n* = 9), Hancock II (*n* = 9), and human pericardium (*n* = 7)). In old mice, statistical differences were found by Kruskal–Wallis and Dunn’s tests between the ATS 3f and Crown cohorts and the other grafted groups, as indicated (* *p* ≤ 0.05, ** *p* ≤ 0.005).

**Figure 3 bioengineering-13-00053-f003:**
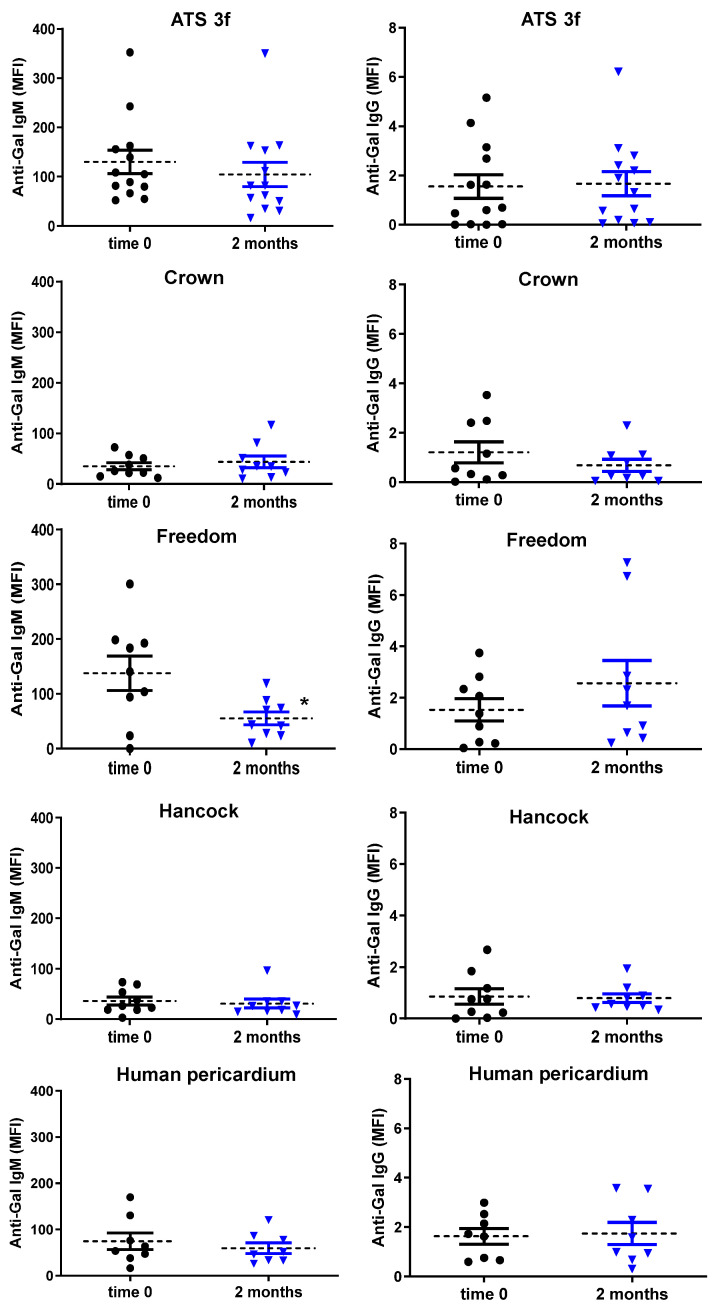
Levels of serum anti-Gal antibodies in old Gal KO mice grafted with BHV tissues. Anti-Gal IgM and IgG levels were measured by flow cytometry of PAEC incubated separately with each of the mouse sera alone or with saturating concentrations of GAS914 to set up the background. Individual determinations (black dots for baseline, blue triangles for 2 months) and the average (dashed line) mean fluorescence intensity (MFI) ± SEM (error bars) of anti-Gal IgM and IgG reactivity after subtracting the background of each determination are shown. It includes, as indicated, all cohorts of old mice implanted with fragments of either ATS 3f (*n* = 13), Crown (*n* = 9), Freedom-Solo or Pericarbon Freedom (Freedom, *n* = 9), Hancock II (Hancock, *n* = 9), and human pericardium (*n* = 8). Statistical differences were only reached for reduced IgM in mice implanted with Freedom relative to baseline (time 0) by paired Student’s *t*-test (* *p* ≤ 0.05).

**Figure 4 bioengineering-13-00053-f004:**
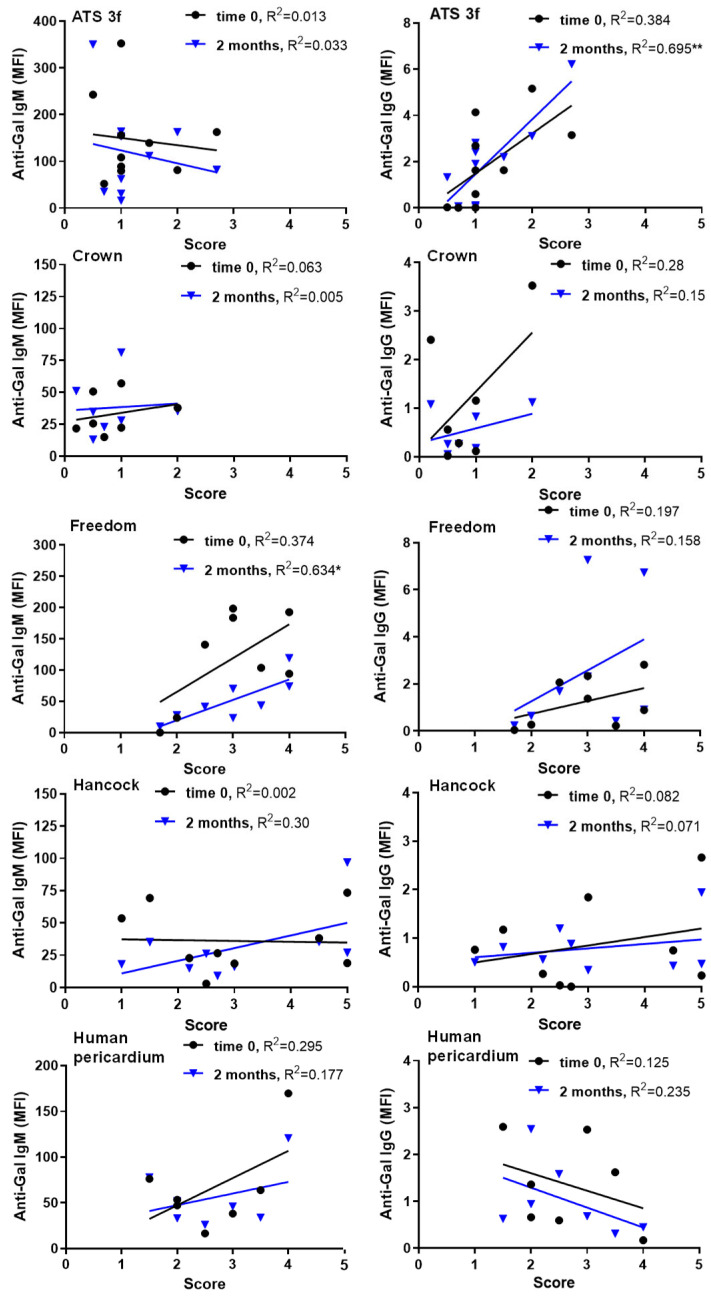
Correlations of serum anti-Gal antibodies with histological scores in old Gal KO mice grafted with BHV tissues. The correlation between the mean fluorescence intensity (MFI) of anti-Gal IgM and IgG reactivity (baseline in black, and 2 months post-implantation in blue) and the histological score was calculated for each cohort of implanted old mice [ATS 3f (*n* = 11), Crown (*n* = 9), Freedom Solo/Pericarbon Freedom (Freedom, *n* = 9), Hancock II (Hancock, *n* = 8), and human pericardium (*n* = 7)]. Each symbol corresponds to an antibody determination associated with a particular histological score (black dots for baseline, blue triangles for 2 months). The coefficient of determination (R^2^) was obtained by Pearson r (* *p* ≤ 0.05, ** *p* ≤ 0.005). It was found to be statistically significant for anti-Gal IgM at 2 months post-implantation of Freedom tissue and for anti-Gal IgG at 2 months post-implantation for ATS 3f tissue. In these two cases, a positive correlation was observed, depicted by an increasing blue line surrounded by blue triangles, with the higher infiltration corresponding to the higher serum antibody levels.

**Table 1 bioengineering-13-00053-t001:** Body weight (g, mean ± SEM) of adult and old Gal KO mice grafted subcutaneously with BHV tissues for 2 months ^1^.

BHV	Adult Mice		Old Mice	
	Baseline	2 months	Baseline	2 Months
ATS 3f	35.6 ± 0.5	34.6 ± 1.1	33.4 ± 0.9	32.6 ± 1.3
Crown	35.7 ± 0.8	33.7 ± 1.3 *	31.8 ± 1.4	25.3 ± 1.5 **
Freedom	32.8 ± 1.5	33.4 ± 1.2	31.8 ± 1.2	27.2 ± 2.3 *
Hancock	36.3 ± 0.8	34.6 ± 0.3	35.3 ± 1.4	31.2 ± 1.8 **
huPeric	36.4 ± 1.3	35.8 ± 1.2	34.7 ± 1.4	30.1 ± 1.4 *

^1^ Adult Gal KO mice were implanted with fragments of either ATS 3f (*n* = 9), Crown (*n* = 9), Freedom Solo (Freedom, *n* = 10), Hancock II (*n* = 7), or human pericardium (huPERIC, *n* = 7), and the body weight (g) was documented at baseline and 2 months post-implantation. Old Gal KO mice also received either ATS 3f (*n* = 13), Crown (*n* = 9), Freedom Solo (Freedom, *n* = 9), Hancock II (*n* = 9), or human pericardium (huPERIC, *n* = 8), and the body weight was measured. The means ± SEM were calculated, and the statistical significance relative to baseline was determined by paired Student’s *t*-test for the various cohorts (* *p* ≤ 0.05; ** *p* ≤ 0.005).

**Table 2 bioengineering-13-00053-t002:** Histological analyses of BHV tissues grafted subcutaneously in adult and old Gal KO mice for 2 months are shown as mean ± SEM of the assigned scores ^1^.

Scores (1–5)	Adult Mice	Old Mice
ATS 3f	2.2 ± 0.31	1.2 ± 0.19 *
Crown	2.0 ± 0.37	0.8 ± 0.18 *
Freedom	2.5 ± 0.51	2.9 ± 0.27
Hancock	3.2 ± 0.38	3.0 ± 0.49
huPeric	4.1 ± 0.17	2.5 ± 0.34 **

^1^ Adult Gal KO mice were implanted with fragments of either ATS 3f (*n* = 8), Crown (*n* = 8), Freedom Solo (Freedom, *n* = 7), Hancock II (*n* = 7), or human pericardium (huPERIC, *n* = 7), and the graft was recovered for the described analyses at 2 months post-implantation. Old Gal KO mice also received either ATS 3f (*n* = 11), Crown (*n* = 9), Freedom Solo (Freedom, *n* = 9), Hancock II (*n* = 9), or human pericardium (huPERIC, *n* = 7), and the graft was equally harvested and analyzed. Statistical significance is shown after comparing the scores corresponding to adult and old Gal KO mice for each BHV tissue by Student’s *t*-test (* *p* ≤ 0.05; ** *p* ≤ 0.005).

## Data Availability

Data are contained within the article and Appendix A.

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
