# Peer review of "Antibody and Cellular Immune Responses in Old α1,3-Galactosyltransferase-Knockout Mice Implanted with Bioprosthetic Heart Valve Tissues"

_bioengineering, 2025, doi:10.3390/bioengineering13010053_

Round 1

Reviewer 1 Report

Comments and Suggestions for Authors

This manuscript addresses an interesting aspect of bioprosthetic heart valve (BHV) failure related to age-dependent immune responses and provides valuable findings utilizing mice model system. Notably, the relationship between cellular immune infiltration and humoral antibody responses to BHV tissues specifically in aged (old) mice is interesting. However I have few questions need to addressed; 

  1. Flow cytometry data is missing. Authors are asked to include the flow data in the manuscript and also share the raw files as supplementary.
  2. In statistical analysis  - 'p' should be always mentioned as in small letter and in italics.
  3. I was wondering that there is no scale bar in the histology images shown in figure 2 a. Authors are asked to include the scale bar.
  4. The use of α1,3-galactosyltransferase-knockout (Gal KO) mice is central, but the rationale for their selection could be stated more explicitly, noting the relevance to immune response studies.

Author Response

This manuscript addresses an interesting aspect of bioprosthetic heart valve (BHV) failure related to age-dependent immune responses and provides valuable findings utilizing mice model system. Notably, the relationship between cellular immune infiltration and humoral antibody responses to BHV tissues specifically in aged (old) mice is interesting. However I have few questions need to addressed; 

  1. Flow cytometry data is missing. Authors are asked to include the flow data in the manuscript and also share the raw files as supplementary.
  2. In statistical analysis  - 'p' should be always mentioned as in small letter and in italics.
  3. I was wondering that there is no scale bar in the histology images shown in figure 2 a. Authors are asked to include the scale bar.
  4. The use of α1,3-galactosyltransferase-knockout (Gal KO) mice is central, but the rationale for their selection could be stated more explicitly, noting the relevance to immune response studies.

Reply 1

Thank you for your comments. A point-by-point answer is provided:

  1. We have chosen to show the results of the flow cytometry work (Fig. 3 and 4) as MFI for each serum sample corresponding to every single mouse at baseline and at two months post-implantation to facilitate comprehension by the readers. The major reason is the high variability of antibody levels and responses in this mouse model. This effect can be observed in Fig. 3 and 4 in which every dot in the graphs represents a serum sample. In addition, this type of data presentation was used in our previous publication Casós et al. Bioengineering 2023 (ref 19) and we consider it is best to be consistent with the form to facilitate the follow up of both publications. Every single dot in figure 3 corresponds to the subtraction of 2 FACs readings without specific gating, one to set up the maximum reactivity of a particular diluted serum in the absence of GAS914, and another with the same diluted serum incubated in the presence of GAS914 to block anti-Gal antibodies to provide specificity (more detail can be found in the reply to reviewer 3). A representation of this study was published as supplementary material in Casós et al. Bioengineering 2023 (ref 19). A detailed explanation of the procedure and representative images of the original data are now provided as new supplementary material.
  2. This modification has been conducted throughout the manuscript.
  3. Unfortunately, it is not possible to add a scale to the images in Figure 2 since these were obtained a few years ago with a camera connected to an optical microscope in the absence of the scale.
  4. We especially thank the reviewer for pointing out this improvement. We have added a sentence in the introduction to further emphasize the value of this animal model.

Reviewer 2 Report

Comments and Suggestions for Authors

The manuscript (ID: bioengineering-3983448) entitled “The immune response to bioprosthetic heart valve tissues in old α1,3-galactosyltransferase-knockout mice” describes a solid piece of scientific work and is in principle be suited for publication in the chosen target journal. However, some additions and improvements are still necessary. The authors are discussing the Galili – antigen. Why don’t they mention it?

Galili, M.R. Clark, S.B. Shohet, J. Buehler & B.A. Macher (1987) Evolutionary relationship between the natural anti-Gal antibody and the Gal alpha 1-3Gal epitope in primates., Proc. Natl. Acad. Sci. U.S.A. 84 (5) 1369-1373. https://doi.org/10.1073/pnas.84.5.1369

Furthermore, tis publication has to be considered in the context of the submitted manuscript:

Veraar Cecilia, Koschutnik Matthias, Nitsche Christian, Laggner Maria, Polak Dominika, Bohle Barbara, Mangold Andreas, Moser Bernhard, Mascherbauer, Julia, Ankersmit Hendrik J. (2021) Inflammatory immune response in recipients of transcatheter aortic valves. 2021/06/01. doi: 10.1016/j.xjon.2021.02.012

The authors mention the Neu5Gc epitope. It would be indeed of highest importance to discuss sialic acids in relation to bioprosthetic heart valve tissues and a potential use of additional experiments.

Breimer ME and Holgersson J (2019) The Structural Complexity and Animal Tissue Distribution of N-Glycolylneuraminic Acid (Neu5Gc)-Terminated Glycans. Implications for Their Immunogenicity in Clinical Xenografting. Front. Mol. Biosci. 6:57. doi: 10.3389/fmolb.2019.00057

Runjie Zhang, Ying Wang, Lei Chen, Ronggen Wang, Chu Li, Xiaoxue Li, Bin Fang, Xueyang Ren, Miaomiao Ruan, Jiying Liu, Qiang Xiong, Lining Zhang, Yong Jin, Manling Zhang, Xiaorui Liu, Lin Li, Qiang Chen, Dengke Pan, Rongfeng Li, David K.C. Cooper, Haiyuan Yang, Yifan Dai (2018) Reducing immunoreactivity of porcine bioprosthetic heart valves by genetically-deleting three major glycan antigens, GGTA1/β4GalNT2/CMAH, Acta Biomaterialia, Volume 72, Pages 196-205. https://doi.org/10.1016/j.actbio.2018.03.055

Schauer R, Kamerling JP. Exploration of the Sialic Acid World. Adv Carbohydr Chem Biochem. 2018;75:1-213. doi: 10.1016/bs.accb.2018.09.001. Epub 2018 Nov 28. PMID: 30509400; PMCID: PMC7112061.

Author Response

Comments and Suggestions for Authors

  1. The manuscript (ID: bioengineering-3983448) entitled “The immune response to bioprosthetic heart valve tissues in old α1,3-galactosyltransferase-knockout mice” describes a solid piece of scientific work and is in principle be suited for publication in the chosen target journal. However, some additions and improvements are still necessary. The authors are discussing the Galili – antigen. Why don’t they mention it?

Galili, M.R. Clark, S.B. Shohet, J. Buehler & B.A. Macher (1987) Evolutionary relationship between the natural anti-Gal antibody and the Gal alpha 1-3Gal epitope in primates., Proc. Natl. Acad. Sci. U.S.A. 84 (5) 1369-1373. https://doi.org/10.1073/pnas.84.5.1369

  1. Furthermore, this publication has to be considered in the context of the submitted manuscript:

Veraar Cecilia, Koschutnik Matthias, Nitsche Christian, Laggner Maria, Polak Dominika, Bohle Barbara, Mangold Andreas, Moser Bernhard, Mascherbauer, Julia, Ankersmit Hendrik J. (2021) Inflammatory immune response in recipients of transcatheter aortic valves. 2021/06/01. doi: 10.1016/j.xjon.2021.02.012

  1. The authors mention the Neu5Gc epitope. It would be indeed of highest importance to discuss sialic acids in relation to bioprosthetic heart valve tissues and a potential use of additional experiments.

Breimer ME and Holgersson J (2019) The Structural Complexity and Animal Tissue Distribution of N-Glycolylneuraminic Acid (Neu5Gc)-Terminated Glycans. Implications for Their Immunogenicity in Clinical Xenografting. Front. Mol. Biosci. 6:57. doi: 10.3389/fmolb.2019.00057

Runjie Zhang, Ying Wang, Lei Chen, Ronggen Wang, Chu Li, Xiaoxue Li, Bin Fang, Xueyang Ren, Miaomiao Ruan, Jiying Liu, Qiang Xiong, Lining Zhang, Yong Jin, Manling Zhang, Xiaorui Liu, Lin Li, Qiang Chen, Dengke Pan, Rongfeng Li, David K.C. Cooper, Haiyuan Yang, Yifan Dai (2018) Reducing immunoreactivity of porcine bioprosthetic heart valves by genetically-deleting three major glycan antigens, GGTA1/β4GalNT2/CMAH, Acta Biomaterialia, Volume 72, Pages 196-205. https://doi.org/10.1016/j.actbio.2018.03.055

Schauer R, Kamerling JP. Exploration of the Sialic Acid World. Adv Carbohydr Chem Biochem. 2018;75:1-213. doi: 10.1016/bs.accb.2018.09.001. Epub 2018 Nov 28. PMID: 30509400; PMCID: PMC7112061.

Reply 2

Thank you for your comments, a point-by-point answer is provided:

  1. We certainly acknowledge the contribution of Dr. Galili to the field as we have worked with the Gal antigen and the Gal KO mouse model for many years. In this respect, we have included a recent reference of his work (number 20) that we think addresses this point as well as another brought by reviewer 1.

Galili U. Antibody production and tolerance to the alpha-gal epitope as models for understanding and preventing the immune response to incompatible ABO carbohydrate antigens and for alpha-gal therapies. Front. Mol. Biosci. 2023, 10:, 209974. doi: 10.3389/fmolb.2023.1209974.

  1. A sentence and the corresponding reference have been included in the discussion relating our results to the work by Veraar et al.. As the TAVI patients described in the selected publication are of old age and show increases in inflammatory markers, we agree that it is particularly suitable to relate this work to the discussion about the positive correlations found for anti-Gal antibodies and cellular immunity. It raises the value of our animal model. Although Gal KO mice show weaker anti-Gal antibody responses than humans, it also reinforces the idea that the lack of increases in serum anti-Gal antibodies does not guarantee that anti-Gal antibodies and complement are not present on the graft.
  2. A short paragraph has been included at the end of the discussion with two of the references you mention. We comment on potential future studies in these models. We also reinforce the concept that the Gal KO mouse may be considered a model for assessing immunity against carbohydrates in general.

Reviewer 3 Report

Comments and Suggestions for Authors

Thank you for submitting this well-designed and scientifically valuable manuscript. The study provides important insights into age-related differences in humoral and cellular immune responses to bioprosthetic heart valve tissues using the Gal KO mouse model. The work is original, relevant, and methodologically sound. Only minor revisions are required to improve clarity and presentation.

Major comments (within Minor Revision scope)
    1.    Flow cytometry methodology
The description of the gating strategy and the role of GAS914 controls could be clarified further. Providing a schematic example or a supplementary figure of the gating workflow would improve transparency and reproducibility.
    2.    Figure legends
Figures 2–4 would benefit from slightly expanded legends. Explicitly listing the sample size (n) within each figure legend and clarifying the meaning of the regression lines in Figure 4 would enhance readability.
    3.    Discussion – mechanistic detail
The Discussion section clearly interprets the findings; however, the manuscript would be strengthened by briefly elaborating on why Freedom/Solo and Hancock II maintain robust immunogenicity in older mice. Offering a more specific mechanistic hypothesis would benefit the reader.

Minor comments
    1.    Several tables (e.g., Tables 3–6) contain the term “mean fluoresce intensity” — this should be corrected to “mean fluorescence intensity (MFI)”.
    2.    Ensure consistent naming of the Freedom/Pericarbon Freedom/Freedom Solo tissue across the text.
    3.    Consider splitting overly long paragraphs in the Discussion to improve readability.

Overall, this is a valuable and well-executed study. After addressing the minor points above, the manuscript will be suitable for publication.

Author Response

Comments and Suggestions for Authors

Thank you for submitting this well-designed and scientifically valuable manuscript. The study provides important insights into age-related differences in humoral and cellular immune responses to bioprosthetic heart valve tissues using the Gal KO mouse model. The work is original, relevant, and methodologically sound. Only minor revisions are required to improve clarity and presentation.

Major comments (within Minor Revision scope)
    1.    Flow cytometry methodology
The description of the gating strategy and the role of GAS914 controls could be clarified further. Providing a schematic example or a supplementary figure of the gating workflow would improve transparency and reproducibility.
    2.    Figure legends
Figures 2–4 would benefit from slightly expanded legends. Explicitly listing the sample size (n) within each figure legend and clarifying the meaning of the regression lines in Figure 4 would enhance readability.
    3.    Discussion – mechanistic detail
The Discussion section clearly interprets the findings; however, the manuscript would be strengthened by briefly elaborating on why Freedom/Solo and Hancock II maintain robust immunogenicity in older mice. Offering a more specific mechanistic hypothesis would benefit the reader.

4. Minor comments
    1.    Several tables (e.g., Tables 3–6) contain the term “mean fluoresce intensity” — this should be corrected to “mean fluorescence intensity (MFI)”.
    2.    Ensure consistent naming of the Freedom/Pericarbon Freedom/Freedom Solo tissue across the text.
    3.    Consider splitting overly long paragraphs in the Discussion to improve readability.

Overall, this is a valuable and well-executed study. After addressing the minor points above, the manuscript will be suitable for publication.

Reply 3

Thank you for your comments, a point-by-point answer is provided:

Major:

  1. To further clarify our methodology as it could be of use to the scientific community, we specify that gating was only applied to alive PAEC in the Materials and Methods section. The calculation of reactivities was performed by subtraction of the MFI generated by the Kaluza analyses. A reference to a supplementary materials section is also included at this level. This supplementary material explains the procedure in detail to facilitate its reproducibility and high level of control (Suppl. Fig 1) and includes representative images of the original analyses (Suppl. Fig 2 and 3). We also recommend visiting the supplementary materials section from Casós et al. Bioengineering 2023 (ref 19) where this methodology was presented for the first time. This type of assay is very sensitive and could be adapted and applied to future studies in similar animal models assessing responses to carbohydrates.

The protocol is explained for further clarification in this reply. The same serum dilution was used for the determination of reactivities by each serum sample. After testing various serum concentrations, we finally ruled out the possibility of conducting both determinations in a single assay and applied a 1% serum for measurements of IgM reactivity and 0,5% serum for IgG reactivity determinations. This 0,5% serum results in very low values of IgG reactivity but prevents interference and saturation of the system by anti-Gal IgM that can compete for the Gal antigen present on the PAEC surface. In the largest experiments (many mouse serum samples assessed simultaneously), IgG and IgM measurements were conducted in separate experiments/days.

For each flow cytometric experiment, all the sera were first diluted 2x in PBS 1% BSA (2% for IgM, 1% for IgG) in a 96-well plate with identified locations and subsequently transferred to an identically distributed 96-well plate containing an equal volume of diluent with GAS914 2x (1 mg/ml) or not for incubation for 20-30 minutes at room temperature. During this time frame, confluent PAEC were harvested and washed twice with PBS 1% BSA by resuspension and centrifugation (one wash in conical tube and one in 96-well plate after equal distribution). After thorough elimination of supernatants by aspiration, PAEC were resuspended with pre-incubated sera and incubated for an additional 30 minutes at 4°C. Standard procedures were next followed and the stained PAEC were passed through the cytometer.  The MFI for each determination was obtained with an overlayed  histogram by Kaluza software as shown in the representative images (Suppl. Fig 2 and 3).

  1. The figure legends of Figure 3 and 4 have been modified to add detail, especially in the case of figure 4, where a detailed explanation of the statistical results has been included. We do agree that this information facilitates the understanding of the results and correlations. We did not find lack of information in Figure 2 legend. Regarding the figures themselves, we would like to keep them in the same format as in our previous publication (Casós et al. 2023) to be consistent.
  2. We elaborated further the explanation of this topic. We think that it is related to the fact that these two tissues (Hancock II and Freedom) are found by the host to be highly immunogenic in first place. Nevertheless, it is of interest to relate it to the responses to other tissues that were reduced in older mice. We speculate that after initiating a stronger cellular immune response to Hancock II and Freedom tissues, older mice may have difficulties in controlling the stronger inflammatory response. In the Freedom tissue it involves the anti-Gal IgM response, which is certainly proinflammatory.
  3. Minor comments: All the recommendations were followed. In particular, the terminology for Pericarbon Freedom and Freedom Solo has been revised throughout the manuscript. We use the term “Freedom” for simplification when mentioning the tissue but continue to use the full terms when it is of interest to identify a specific BHV (i.e. in the abstract, figure legends for providing methodological detail or the discussion).